# Exercise across the Lung Cancer Care Continuum: An Overview of Systematic Reviews

**DOI:** 10.3390/jcm12051871

**Published:** 2023-02-27

**Authors:** Lara Edbrooke, Amy Bowman, Catherine L. Granger, Nicola Burgess, Shaza Abo, Bronwen Connolly, Linda Denehy

**Affiliations:** 1Department of Physiotherapy, The University of Melbourne, Melbourne, VIC 3010, Australia; 2Department of Health Services Research, The Peter MacCallum Cancer Centre, Melbourne, VIC 3000, Australia; 3Department of Physiotherapy, The Peter MacCallum Cancer Centre, Melbourne, VIC 3000, Australia; 4Department of Physiotherapy, The Royal Melbourne Hospital, Melbourne, VIC 3050, Australia; 5Department of Physiotherapy, Austin Health, Melbourne, VIC 3084, Australia; 6Wellcome-Wolfson Institute for Experimental Medicine, Queen’s University Belfast, Belfast BT7 1NN, UK

**Keywords:** lung cancer, exercise, rehabilitation, overview of reviews

## Abstract

Background: Growing evidence supports exercise for people with lung cancer. This overview aimed to summarise exercise intervention efficacy and safety across the care continuum. Methods: Eight databases (including Cochrane and Medline) were searched (inception—February 2022) for systematic reviews of RCTs/quasi-RCTs. Eligibility: population—adults with lung cancer; intervention: exercise (e.g., aerobic, resistance) +/− non-exercise (e.g., nutrition); comparator: usual care/non-exercise; primary outcomes: exercise capacity, physical function, health-related quality of life (HRQoL) and post-operative complications. Duplicate, independent title/abstract and full-text screening, data extraction and quality ratings (AMSTAR-2) were completed. Results: Thirty systematic reviews involving between 157 and 2109 participants (n = 6440 total) were included. Most reviews (n = 28) involved surgical participants. Twenty-five reviews performed meta-analyses. The review quality was commonly rated critically low (n = 22) or low (n = 7). Reviews commonly included combinations of aerobic, resistance and/or respiratory exercise interventions. Pre-operative meta-analyses demonstrated that exercise reduces post-operative complications (n = 4/7) and improves exercise capacity (n = 6/6), whilst HRQoL findings were non-significant (n = 3/3). Post-operative meta-analyses reported significant improvements in exercise capacity (n = 2/3) and muscle strength (n = 1/1) and non-significant HRQoL changes (n = 8/10). Interventions delivered to mixed surgical and non-surgical populations improved exercise capacity (n = 3/4), muscle strength (n = 2/2) and HRQoL (n = 3). Meta-analyses of interventions in non-surgical populations demonstrated inconsistent findings. Adverse event rates were low, however, few reviews reported on safety. Conclusions: A large body of evidence supports lung cancer exercise interventions to reduce complications and improve exercise capacity in pre- and post-operative populations. Additional higher-quality research is needed, particularly in the non-surgical population, including subgroup analyses of exercise type and setting.

## 1. Introduction

Globally, lung cancer accounted for over 2.2 million (11.4%) incident cancers in 2020 [1]. The majority of diagnoses, 54% in the United States, occur once the disease has metastasized and is considered incurable [2]. In Australia, although lung cancer incidence is reducing, 90,000 new cases are predicted between 2040–2044 due to the ageing population. This incidence is the third and fourth highest of all tumour types for males and females, respectively [3]. Significantly, the number of people dying from lung cancer is predicted to decline in both sexes between 2020 and 2040–2044 [3]. These factors together will mean a growing number of people with lung cancer.

As an area of research, the evaluation of the effects of exercise on people with lung cancer has developed more slowly than in other areas. Study interventions commonly involve combinations of aerobic, resistance and respiratory training (some include inspiratory muscle training). Interventions aim to reduce the risk of post-operative pulmonary complications and aid recovery following surgery. In inoperable populations, the focus of interventions is on reducing symptom burden, commonly fatigue, pain and dyspnoea, and preventing a decline in physical function and health-related quality of life [4]. The first systematic review was published by Granger and colleagues in 2011 and included two randomised controlled trials (RCTs), nine case series and two cohort studies [5]. In contrast, a search of the International prospective register of systematic reviews (PROSPERO) in August 2022 using the terms ‘lung cancer’ AND (‘exercise’ OR ‘rehabilitation’) retrieved 145 registered systematic reviews, highlighting the rapid development of research in this area. The increasing volume and strength of evidence, media attention and an increasing number of lung cancer survivors place increasing demands on limited-exercise oncology services [6]. Evidence-based strategies to direct services to patients who are likely to benefit the most are essential. Overviews of reviews to synthesise the growing evidence base are now required. In people having surgery for lung cancer, Zhou and colleagues report an overview of systematic reviews, published prior to October 2019, of lung cancer exercise interventions delivered during the perioperative period [7]. Findings included low-quality evidence that pre-operative programs reduce post-operative pulmonary complications and hospital length of stay and increase exercise capacity and pulmonary function. Moderate to high-quality evidence supported post-operative programs in increasing exercise capacity and muscle strength. There was very low to low-quality evidence that programs improved health-related quality of life (HRQoL) and reduced dyspnoea [7]. Exercise adherence varies widely and further research to identify enablers of participation is required; an adherence of between 9–125% is reported by pre-and rehabilitation lung cancer studies which include a home-based component [8].

This overview of systematic reviews aimed to synthesise findings from and evaluate the quality of the current systematic review evidence on the efficacy and safety of exercise for people with both operable and inoperable lung cancer, delivered across the care continuum. The secondary aims were to report subgroup meta-analysis findings according to the type of exercise intervention/s and delivery settings, where possible.

## 2. Materials and Methods

This overview of reviews was guided by the Cochrane Handbook of Systematic Reviews of Interventions [9] and is reported according to the Preferred Reporting Items for Systematic Review and Meta-Analysis (PRISMA2020) guidelines [10]. The protocol was registered prospectively on the PROSPERO database (CRD42015001068 https://www.crd.york.ac.uk/prospero/display_record.php?ID=CRD42021257938 (registered on 6 July 2021)).

### 2.1. Eligibility Criteria

The inclusion criteria for systematic reviews were as follows: Population—Patients (≥18 yo) diagnosed with lung cancer (non-small cell or small cell); Intervention—Any supervised or unsupervised exercise interventions delivered alone or in combination with any non-exercise interventions (e.g., nutrition, symptom management, psychological support); Comparator—Usual care or a non-exercise intervention; Context—Any setting (hospital, community or home); Outcomes—At least one health-related outcome. The primary outcomes of interest were exercise capacity, physical function, HRQoL and postoperative pulmonary complications (PPCs). Appendix A in the Appendix A provides further details.

Systematic reviews, with or without a meta-analysis, including only RCTs or quasi-randomised controlled trials (qRCTs), were included. Systematic reviews including findings from other study designs were included if RCT or qRCT findings were reported separately. Abstract-only citations (e.g., conference proceedings) and narrative or non-systematic reviews were excluded. Additionally, systematic reviews where the study population included mixed cancer groups with <50% lung cancer, or mixed cancer groups with lung cancer findings not reported separately were excluded. Only systematic reviews available in English were included.

### 2.2. Literature Search

A comprehensive literature search of eight databases was performed; the Cochrane Systematic Review Database, the Database of Abstracts of Reviews of Effectiveness (DARE), Cochrane Central Register of Controlled Trials (CENTRAL) (The Cochrane Library), Ovid SP MEDLINE, Ovid SP EMBASE, SPORTDiscus and CINAHL via EBSCO host and PEDro from inception until the 18 May 2021 and updated on 21 February 2022. The search string was developed in consultation with content specialists and a research librarian from the University of Melbourne, using the medical subject headings (MeSH) dictionary in MEDLINE to identify key terms and was adapted for use in CINAHL, SPORTDiscus, PEDro, CENTRAL and EMBASE. The full search strategy for each database is provided in the Appendix A (Appendix A).

Two researchers (LE and AB) independently screened the titles and abstracts of all articles retrieved and conducted a full-text review of all articles considered potentially relevant. Consensus between the two researchers was used to resolve any disagreements and a third researcher (LD) was available if a consensus could not be reached. Reference management software Covidence was used for managing all retrieved records [11]. Additional potentially relevant articles were identified by screening the reference lists of all included full-text articles.

### 2.3. Data Extraction and Quality Assessment

Two researchers (AB and NB) independently extracted data using a standardised form developed prior to database searching. A third researcher (LE) resolved any discrepancies in the data extraction. Data were collected for (1) review characteristics, including search dates, language restrictions, synthesis and quality appraisal methods; (2) primary study designs; (3) review population, intervention/s, comparator/s and outcomes; and (4) meta-analysis results for the overview primary outcomes. The authors of the included systematic reviews were contacted by email where discrepancies were unable to be resolved by the researchers or where data were not reported. Grades of Recommendations, Assessment, Development and Evaluation (GRADE) ratings, reported in the included systematic reviews, were extracted. Where GRADE was not reported for an overview primary outcome (e.g., physical function in the ‘During and post-treatment (non-surgical)’ included systematic reviews), two researchers (LE and SA) independently performed GRADE evidence certainty assessments, with any disagreements resolved by a third researcher (LD) [12].

The overall quality of reporting and methodological rigour of included systematic reviews was critically appraised by two researchers (LE and AB) independently, using the PRISMA-2020 item and abstracts [13] checklists and the Assessment of Multiple Systematic Reviews (AMSTAR-2) [14]. PRISMA-2020 is a 27-item (some with sub-items) checklist covering 7 areas (title, abstract, introduction, methods, results, discussion and other information (e.g., protocol registration, funding support). In case of disagreement, a consensus was reached by discussion and a third researcher (LD) was available if consensus could not be reached. The AMSTAR-2 contains 16 items, 7 of which are regarded ‘critical’ (protocol registration, appropriate literature search, excluded study justification, individual study risk of bias, methodological appropriateness of meta-analyses, risk of bias considered in result interpretation and publication bias likelihood/impact). In line with AMSTAR-2 scoring guidelines, confidence in the findings of the systematic reviews was rated ‘critically low’ if there was more than one critical flaw, ‘low’ for systematic reviews with one critical flaw, ‘moderate’ for those with no critical flaws but more than one non-critical weakness and ‘high’ for systematic reviews with one non-critical weakness, or with none [14].

### 2.4. Synthesis of Results

Findings from meta-analysed data included in the overview were synthesised narratively at the review level for each overview primary outcome. Included systematic review characteristics, findings, risk of bias and GRADE assessments were reported in subgroups according to the stage on the cancer treatment continuum that the interventions were delivered: (1) pre-treatment only (surgical); (2) post-treatment only (surgical); (3) pre- and post-treatment (surgical); (4) pre, during and/or post-treatments (mixed surgical and non-surgical) and; (5) during and post-treatments (non-surgical only).

The primary study overlap between the included systematic reviews was reviewed by authors (LE and AB) and visually mapped using a citation matrix. In the event of updated systematic reviews or systematic reviews that included identical studies and addressed the same research question, the more recent review was included. A citation matrix was created, and the corrected covered area (CCA) was calculated to assess the degree of primary study overlap. A CCA within the range of 0–5% indicates slight overlap, 6–10% indicates a moderate overlap, 11–15% indicates a high overlap and >15% indicates a very high amount of overlap [15].

### 2.5. Protocol Deviations

Two deviations from the prospectively registered protocol were made: the addition of a language restriction to English only, and the inclusion of electrical stimulation as an additional exercise intervention within the eligibility criteria.

## 3. Results

### 3.1. Search Results

The search retrieved 5068 articles, and 4114 were screened for eligibility following the removal of duplicates (refer to Figure 1 PRISMA flow diagram for further details). Ninety-seven articles were retrieved for full-text review. Details of excluded articles at full-text screening and reasons for exclusion are provided in the Appendix A (Appendix A). Thirty systematic reviews (SRs) involving 6440 participants were included in the overview, of which 25 included synthesised findings in at least one meta-analysis.

**Figure 1 jcm-12-01871-f001:**
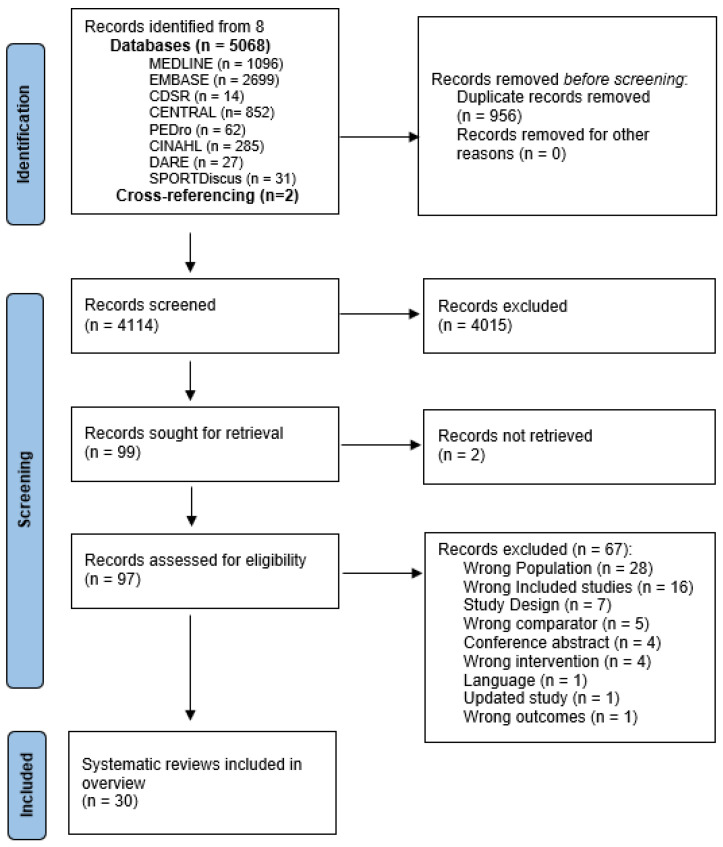
Overview flowchart.

### 3.2. Methodological Rigour and Quality of Reporting

AMSTAR-2 ratings were ‘critically low’ in 22 (73.3%), ‘low’ in 7 (23.3%) and ‘moderate’ in 1 (3.3%) of the included SRs. All SRs used an appropriate method for assessing the risk of bias of individual studies. Nine SRs (30%) provided a list with justification for excluded studies and ten (33.3%) accounted for individual study risk of bias in interpreting findings. Twelve SRs (40%) provided justification for their study design inclusion criteria. Further details are provided in the Appendix A (Appendix A).

All SRs met PRISMA 2020 guideline items relating to reporting a rationale for the review and flow diagram of included studies; outlining included study characteristics; and interpreting their findings. Twenty (67%) SRs did not provide citations and justification for the exclusion of studies and nineteen (63%) did not assess certainty of the evidence. The Appendix A provides further details (Appendix A).

### 3.3. Characteristics of Included Studies

#### 3.3.1. Participants, Interventions and Outcomes

Table 1 provides details of included SRs, including participants, interventions, comparators, outcomes, synthesis and quality appraisal methods. A narrative summary according to stage on the treatment continuum is provided below.

**Table 1 jcm-12-01871-t001:** Characteristics of included reviews.

Author Year, SR Focus and Search Dates	Primary Studies (Number; Number Pre, During or Post Treatment (if Mixed))	Participant Characteristics (n, Treatment Type (if Mixed))	Type (n NSCLC/SCLC/Mesothelioma); Stage (TNM Classification)	Exercise Intervention Types, Prescription Ranges (F.I.T.T.), Supervised and/or Unsupervised)	Additional Intervention/s (Number of Studies, Combined with What?)	Comparator	Outcomes (* Primary, if Reported)	Languages Searched
Pre-treatment only (surgical)								
Cavalheri 2017 [4] Lung cancer exercise training. Inception—Nov 2016	5 RCTs	203, surgical	NSCLC (one study (n = 19) did not specify type); Stage: 100 I–IIIB, 103 NR	Type: aerobic, resistance, respiratory (including IMT), education. FITT: 3× per day for 1 week to 5× per week for 4 weeks. Intensity: reported in 2 studies (70–80% max work rate). Supervision: NR.	1× trial—both groups received education re energy conservation techniques, relaxation and stress management and focus on nutrition. ×2 trials described as ‘pulmonary rehabilitation’ components in addition to exercise NR	Non-exercise training or usual care	PPC *, days with ICC post-op *, LOS, fatigue, exercise capacity, pulmonary function, postoperative mortality, dyspnoea	No language restriction
Steffens 2018 [16] Cancer (any type) exercise interventions Inception—Nov 2016	5 RCTs + 1 qRCT (of 13 included studies)	434, surgical	Type and stage NR	Type: aerobic, resistance, respiratory (including IMT). FITT: 20–60 min, 3× daily for 1 week to 5× weekly for 2 weeks. 1× NR duration, 1× NR dosage. Supervision: NR.	Exercise only	Usual care	Post-operative complications, LOS	No language restriction
Treanor 2018 [17] Prehabilitation interventions for newly diagnosed cancer. Inception—Apr 2017	7 RCTs (of 18 included studies)	449, surgical	197 NSCLC, 252 NR; Stage: 36 I-IIIa, 88 NR.	Type: aerobic, resistance, respiratory. FITT: 1–3× per day to 1 day per week for 1–3 weeks. 1× NR dosage. Intensity: reported by 1 study (Licker 2016) 80–100% peak work rate. Supervision: combined supervised hospital and home based.	×3 trials described as ‘pulmonary rehabilitation’—components in addition to exercise NR	Usual care (×1 study included daily walking advice)	LOS, post-op complications, time intubated, pulmonary function, QoL and feelings of hope and power	English
Rosero 2018 [18] Physical exercise for people with NCSLC. Jan 1970–Feb 2018	10 RCTs	676, surgical	NSCLC; Stage: 625 I–II, 51 III-IV	Type: aerobic, resistance, respiratory (including IMT), PNF, stretching FITT: 20–60 min, 1–2× per day to 3–7 days per week, 1–4 weeks. Intensity: ×4 studies—70–100% PWC, ×6 NR. Supervision: supervised.	Psychological educational guidance (×1 trial), pharmacotherapy (×1 trial)	Usual care	Exercise capacity *, lung function, dyspnoea, fatigue, PPC, post-op days in hospital, HRQoL, RPE	English
Li 2019 [19] Exercise therapy effects on surgical outcomes in lung cancer with or without COPD. Inception–June 2017	7 RCTs + 1 prospective cohort with retrospective control	404, surgical	NSCLC; Stage: 25 I–III, 130 I–IV, 40 NR	Type: aerobic (including HIIT), resistance, respiratory. FITT: 15–30 min, 1–3× per day, 5–7× per week over 1–4 weeks. Intensity: NR. Supervision: combined supervised hospital and home-based.	Pharmacotherapy (×1 trial)	Usual care (including chest physiotherapy and breathing exercises)	PPC *, duration of ICC *, LOS *, pulmonary function, exercise capacity, dyspnoea	No language restriction
Pu 2021 [20] Impact of respiratory exercises for people with NSCLC. Inception -Mar 2021	10 RCTs	768, surgical	NSCLC; Stage NR.	Type: aerobic, respiratory (including IMT). FITT: 3 days–4 weeks IMT: between 10–20 min, 2–4× per day. AT + RT:15–30 min, daily to 3× per week. Intensity: modified Borg ‘light’—7, Borg 13–16, 70% CPET max score. Supervision: NR	Exercise only	Standard of care	PPC *, LOS *, HRQoL, mortality, surgical complications, lung function and exercise capacity	English
Gravier 2021 [21] Effects of exercise training on post-operative complications and other outcomes. Inception—Dec 2020	14 RCTs	791, surgical	Type NR (eligibility states NSCLC); Stage: 60 IA-IIIB, 73 NR	Type: aerobic, resistance, respiratory (including IMT), stretching, balance. FITT: 1–8 weeks, ×2–3/week to ×3/day. Intensity: 50–100% Wpeak, 13–16 Borg RPE. Supervision: supervised IP or OP, ×2 studies unsupervised home-based.	Pulmonary rehabilitation (×1 trial), education (×2 trials), psychology (×1 trial), psychology + nutrition (×1 trial)	Usual care or only chest physiotherapy or education	Post-op complications (POC) *, 30-day mortality *, LOS, exercise capacity, respiratory function, QoL, anxiety, depression, program adherence, program completion, AEs	No language restriction
de Oliveira Vacchi 2022 [22] Effects of IMT +/− rehabilitation on functional and pulmonary capacity. Inception—Nov 2019	6 RCTs	219, surgical	Type and stage NR	Type: aerobic, resistance, respiratory (including IMT). FITT: ×2/day to ×5/week, 1–4 weeks. Intensity: IMT: 15–20% MIP, increased by 5–10% to max 60%. AT: ‘high’ ×2 studies, 80% max load × 2 studies. Supervision: NR.	Exercise only	Usual care or only chest physio or education	Functional capacity, pulmonary function, quality of life, post-operative complications, LOS, mortality	No language restriction
Post-treatment only (surgical)								
Li 2017 [23] Efficacy of exercise of training following lung resection surgery. Inception -Feb 2017	6 RCTs	438 l, surgical	NSCLC 370, mixed 68 (NSCLC + metastatic tumour + other type); Stage NR	Type: aerobic, resistance, respiratory (including IMT). FITT: 2× daily to 2× per week, 5–60 min for 2–20 weeks. Intensity: 60–95% MHR. Supervision: NR.	Exercise only	Usual care or standard physiotherapy treatment	Quality of life *, exercise capacity, physical activity, lung function, POC, PPC, muscle strength, symptoms (pain, dyspnoea)	No language restriction
Sommer 2018 [24] Post-surgical rehabilitation for lung cancer patients. Inception—Feb 2016	4 RCTs (within 1 year of surgery)	262, surgical	NSCLC NR (states ‘mainly’); Stage NR (‘evenly distributed between groups’)	Type: aerobic, resistance, respiratory (IMT), whole body vibration. FITT: 1× weekly for 10 weeks to 3× weekly for 12–20 weeks. Initiated between 5 days–8 weeks following lung resection. Intensity: 60–95% MHR, 70% WMax, RPE 11–16. Supervision: supervised outpatient. 1× study also included inpatient and unsupervised.	Exercise only	Usual care (including home exercise education), general information and discouraged to improve exercise tolerance.	Exercise capacity *, HRQoL	English, one of the Scandinavian languages, or German
Cavalheri 2019 [25] Post-operative rehabilitation for lung cancer patients. Inception—Feb 2019	8 RCTs (within 1 year of surgery)	450, surgical	NSCLC; Stage: 72 I, 33 II, 13 III, 6 IV, NR 350.	Type: aerobic, resistance, respiratory (including IMT), balance. FITT: 5–60 min, 2× day–5× per week over 4–20 weeks. Intensity: 60–90% HRR, 3–4 Borg dyspnoea, 13–15 RPE, 50–80% Wmax, 80–95% max HR, 80% 6 MWT speed. Supervision: supervised inpatient and outpatient +/− home-based component.	Exercise group had pain treatment adjusted by anaesthiologist (×1 trial). Nurse counselling, up to 3× 60 min (×1 trial)	Usual care	Exercise capacity *, safety *, pulmonary function, HRQoL, muscle strength, anxiety and depression	No language restriction
Larsen 2019 [26] Respiratory physiotherapy following lung resection. Inception—Mar 2017	11 RCTs and 2 qRCTs (5RCT and 1qRCT included non-LC).	1280 (618 LC), surgical	Type and stage NR.	Type: aerobic, resistance, respiratory. FITT: 5–30 mins, 2× day, for up to 5 days. Intensity: NR. Supervision: supervised inpatient or NR.	Exercise / respiratory PT only	Standard care (Included chest PT in 2× studies)	Mortality, PPC, LOS	No language restriction
Pre and post-treatment (surgical)								
Rueda 2011 [27] Non-invasive interventions to improve wellbeing for people with lung cancer. Inception—Feb 2011	2 RCTs (1× pre- and 1× post-op) of 15 included studies	157; pre (104) and post-operative (53)	NCSLC; Stage: 104 IA–IIIA, 53 NR	Type: aerobic, resistance, respiratory FITT: Pre-op: 7–10 days aerobic/resistance (1× per day) + respiratory (10× per day). Post-op: inpatient aerobic/resistance ×2 per day for 1–5 days—followed by home exercise program. Intensity: NR Supervision: supervised inpatient, unsupervised/home visits home program.	Exercise only	No exercise training	Exercise capacity, Quality of life, hope (Hearth Hope Index) and power (PKPCT), quadriceps strength	NR
Mainini 2016 [28] Pre and post operative physical exercise interventions for patients with NSCLC. May 2013–May 2016	6 RCTs (1× pre- and 5× post-op)	414; pre (40), and post-operative (374),	NSCLC; Stage NR	Type: aerobic, resistance, respiratory (including IMT), balance FITT: Pre-op: aerobic and respiratory, 5× weekly for 3 weeks. Post-op: aerobic, resistance, respiratory (including IMT), balance, range: daily to 1× weekly, 5 days–20 weeks. Intensity: 60–70% CPET work max or 70–80% max 6 MWT speed, RPE 11–16, 60–90% HRR, 80–95% MHR. Supervision: pre-op supervised outpatient, post-op supervised and unsupervised.	Exercise combined with dyspnoea management—both groups 3 × 1 h post-op nurse counselling (×1 trial)	Usual care	Exercise capacity, HRQoL, and lung function (FEV1, FVC and DLCO)	English, French, Italian, Portuguese and Spanish
Wang 2018 [29] Preoperative breathing exercises for patients with operable lung cancer. Inception—Dec 2017	16 RCTs (8 pre-, 4 periop, 4 post-op)	1234	166 NSCLC, 1068 NR; Stage: 398 Ia-IIIb (not receiving radio/chemotherapy), 836 NR.	Type: aerobic (including HIIT), resistance, respiratory (including IMT), balance FITT: 2×day–5× per week over 4–20 weeks. Intensity: ×1 HIIT (Huang 2017), ×1 20–60% MIP, ×7 NR. Supervision: NR.	Exercise only	NR	Pulmonary function *, PPCs *, LOS and exercise capacity	English and Chinese
Himbert 2020 [30] Intervention effects on pulmonary and physical function pre and post, lung cancer surgery. Jan 1946–Mar 2020	22 RCTs (11 pre-, 11 post-op)	893; 560 pre, 333 post-operative (only reported from 5RCTs)	674 NSCLC; Stage: 151 I–IIIa, 742 NR.	Type: aerobic (including HIIT), resistance, respiratory (including IMT), balance FITT: Pre-surgery: 10–60 min, 1–7 days a week over 5 days–8 weeks, moderate-high intensity. Combination of inpatient, outpatient, supervised and home-based. Post-surgery: 5–60 min, 1–7 days a week over 1–20 weeks, moderate-high intensity. Supervised (6 studies), home-based (1 study), both supervised and unsupervised (2 studies).	Exercise only	Usual care	Pulmonary (lung volume and capacity) and physical function (cardiorespiratory fitness, functional capacity, physical performance) domains	English
Machado 2021 [31] Effects of exercise pre and post lung cancer surgery on HRQoL. Inception—Mar 2021	10 RCTs (1 pre, 9 post-op (4 during adjuvant chemo))	651; pre (22), 629 post-op (140 received adjuvant Rx, mainly chemotherapy)	NSCLC; Stage: 454 I-II, 71 NR	Type: aerobic (including HIIT), resistance, respiratory (including IMT), balance, mobility and stretching. FITT: 5–60 mins/session, ×2–3/week (centre-based) and ×5/week (home-based), 4 weeks (pre-op), 6–20 weeks (post-op). Intensity: HIIT (80% Wpeak or 85–100% HRmax) ×5 studies, continuous ×6 studies (light ×1, mod ×5). Supervision: largely supervised.	Relaxation (×1 trial)	Usual care with no exercise training, general PA advice	HRQoL *, fatigue	English
Xu 2022 [32] Effects of exercise pre and post lung cancer surgery on PPCs and LOS. Inception—Jun 2021	23 RCTs (12 pre-, 10 post- and 1 peri-op)	2068; (809 pre-, 1189 post-, 70 peri-operative)	1054 NSCLC, 1014 NR; Stage: 140 I-II, 343 I-III, 164 I-IIIA, 60 I-IIIB, 749 I-IV, 612 NR.	Type: aerobic (including HIIT), resistance, respiratory (including IMT), balance, mobility and shoulder ROM. FITT: Pre-op: ×2/week–×4/day (IMT). Duration: 1–4 weeks, 20–40 mins/session. Post-op: ×1/day-hourly, during I/P admission–12 weeks, 5–30 min. Peri-op: ×2/day, during I/P admission. Intensity: IMT: 15–60% MIP, aerobic 60–80% Wpeak/60–90% HRR, Borg 13–16/11–13, <6. Supervision: Pre-op—largely supervised or supervised/home unsupervised. Post-op—largely supervised. Periop—supervised.	Nutrition and psychosocial (relaxation)—×1 trial	Usual care	Hospital LOS *, post-operative pulmonary complications *, post-operative complications, chest tube duration, mortality	English
Pre, during and/or post-treatment (surgical and non-surgical)								
Hsieh 2017 [33] Supportive care interventions on depressive symptoms for people with lung cancer. Inception—Sept 2015	3 RCTs (1× post-treatment, 2× NR) of 12 included studies	187, treatment type NR	24 NSCLC early stage; 163 type/stage NR	Type: ‘pulmonary rehabilitation’, aerobic, respiratory (including IMT), stretching. FITT: 3–5× per week, 30–40 min/session for 4–12 weeks. Intensity: NR. Supervision: supervised and unsupervised home based.	Exercise only	Standard care	Depressive symptoms *	English or Chinese
Papadopolous 2018 [34] Effects of nonpharmacologic interventions on sleep quality of people with lung cancer. Inception—Dec 2016	3×RCT, 2×qRCT (mixed post-op and during chemo and/or RT)	364 (109 post op, 95 chemo, 4 RT, 3 chemoRT, 153 NR)	Type NR; Stage 72 I, 24 II, 48 III, 149 IV, 71 NR	Type: aerobic, resistance, respiratory, Tai Chi FITT: ×2/day–×1/week, 30–45 min/session. ×2 inpatient, during inpatient admission only—12 weeks. Intensity: NR. Supervision: ×2 home based, ×1 supervised and home-based, ×1 during inpatient admission supervision NR in 3 studies.	15–20 min/week behavioural support sessions (×1 trial)	Usual care or wait-list	Sleep quality *	No language restriction (sensitivity analysis excluded non-English)
Liu 2019 [35] Respiratory exercise effects on exercise capacity and mental health in people with lung cancer. Inception—Apr 2018	15 RCTs (2 pre-op, 2 pre and post-op, 5 post-op, 3 during non-surgical Rx (2 chemotherapy, 1 chemoradiotherapy), 3 unknown)	930 (532 surgery, 149 surgery/chemo/radiotherapy, 20 chemo/radiotherapy, 49 chemo, 180 NR)	Type: 149 NSCLC/SCLC/mesothelioma, 69 NSCLC/SCLC, 82 NSCLC, 630 NR. Stage: 342 I -IV, 588 NR	Type: respiratory (including IMT), aerobic (including HIIT) and resistance. FITT: 15 minutes–3 h, 2–3× per day to 3–5× per week for 1–12 weeks. Intensity NR. Supervision: unsupervised and supervised (inpatient and outpatient).	Oxygen therapy, aerosol inflation (×1 trial), counselling, teaching coping and adaption strategies (×1 trial), acupressure, modified swallow technique and cough easing techniques (×1 trial), education (×1 trial), goal setting (×1 trial)	Conventional care or no treatment	Dyspnea *, exercise capacity *, anxiety, depression,	English and Chinese
Singh 2020 [36] Adverse events, feasibility and effectiveness of lung cancer exercise. Inception—May 2020	32 RCTs (27 pre-op, 5 post-op)	2109 (1695 pre-op, 414 post-op). 8 RCTs during, 17 RCTs post and 7 RCTs during or post adjuvant chemotherapy	NSCLC 1351, NSCLC + SCLC 758; Stage: 137 I/II, 446 III/IV and 1526 mixed.	Type: aerobic, resistance, respiratory, Tai Chi. FITT: Session duration/frequency/intensity: NR. Program duration 1–20 weeks (21 studies < 12 weeks). Supervision: supervised (27 RCTs) and unsupervised (5 RCTs).	Exercise only	Non-exercise control or usual care	Safety (adverse events), feasibility (recruitment, retention and exercise adherence) and health-related outcomes (quality of life, exercise capacity, fatigue, strength, anxiety, depression, sleep, lung function, dyspnoea, pain and hospital LOS)	English
Ma 2020 [37] Effects of exercise on quality of life for people with lung cancer. Inception—Sept 2019	16 RCTs (3× preoperative, 7× post-operative, 6× during and following non-surgical Rx)	758 participants (106 pre-op, 361 post op, 291 advanced/treatment NR)	704 NSCLC, 49 NSCLC or SCLC, 5 SCLC; Stage 165 I, 79 II, 81 III, 81 IV; 37 I-II, 155 III-IV, 71 I-IV, 6 recurrent, 5 no malignancy, 38 NR.	Type: aerobic, resistance, respiratory FITT: 5–60 min, 1–7 x per week, 1–20 weeks. Supervision: supervised (12 RCTs), unsupervised (1 RCT) and both (3 RCTs)—home-based and outpatient.	Exercise only	Usual care/ no intervention (3× trials included chest physiotherapy/conventional physiotherapy)	QoL *	English
Yang 2020 [38] Home-based exercise effects on exercise capacity, symptoms, and quality of life in people with lung cancer. Inception—Dec 2018	7 RCTs (4× post-op, 3 NR)	559 (175 post-op, 88 post-op + during adjuvant chemotherapy, 296 NR)	175 NSCLC, 384 NR; Stage NR	Type: aerobic, resistance, balance, Tai Chi, Chinese Medicine Baduanjin FITT: 5–60 min, 2× daily to 3× per week, for 6 weeks to 3 months. Intensity: light (1 RCT), 60–80% HR max (2 RCTs), NR 4 RCTs. Supervision: all home-based, level of supervision NR.	Exercise only	Routine guidance	Exercise capacity, cancer-related fatigue, pain, insomnia, appetite loss, coughing, anxiety, depression and HRQoL	English and Chinese
Codima 2021 [39] Exercise for symptom management and quality of life for people with lung cancer. Jan 1998–Jan 2019	10 RCTs (4× pre-op, 3× post-op, 2× during chemotherapy or targeted therapy, 1× mixed)	835 (225 pre-op, 374 post-op, 120 systemic therapy, 116 diverse/no Rx)	657 NSCLC, 178 NR; Stage: 259 I-IIIa, 24 IIIa-IV, 434 I-IV, 78 I-II, 40 NR	Type: aerobic, resistance, respiratory, Tai Chi, stretching. FITT: 30–60 min, ×1/day–×1 per week for 1 to 20 weeks. Intensity: 60–100% HRmax or VO_2 peak_, Borg 11–16 (5 RCTs). Supervision: Supervised (7 RCTs), partially supervised (2 RCTs), and unsupervised (1 RCT). Inpatient, outpatient and home-based.	Group and individual counselling (×1 trial), relaxation and counselling (×1 trial)	Usual care	HRQoL and symptoms	English
Ma 2021 [40] Nonpharmacological interventions for cancer related fatigue for people with lung cancer. Inception—Jun 2020	6 RCTs (1× post-op, 1× post-op/post chemo, 2× during chemo, 1× during targeted therapy, 1× NR) of 18 included studies	364 (72 post-op, 164 during treatment incl target therapy and chemotherapy, 17 curative intent treatment, 111 NR)	Type: NR; Stage: 163 I-IV, 17 I-III, 135 III-IV, 49 IV.	Type: aerobic, resistance, Tai Chi and electrical muscle stimulation. FITT: 30–60 min, 1–7× per week over 6–12 weeks. Intensity: 1×high, others NR. Supervision: 1×RCT home based, 1× RCT supervised, others: setting/supervision NR.	Exercise only	Usual Care, ×1 RCT included low impact exercise (ROM, stretching and breathing) (×1 trial)	Fatigue *	English
Heredia-Ciuro 2021 [41] Effects of HIIT in lung cancer survivors. Inception—Mar 2021	3 RCTs, 2 pilot RCTs, 3 prospective randomised open, blinded end-point (PROBE) studies (3× pre-op, 2× post-op, 1× post-op/post chemo, 1× during RT, 1× during targeted therapy)	305 (151 pre-op, 98 post-op, 17 post-op/post chemo, 15 during RT, 24 during targeted therapy)	Type: 281 NSCLC, 24 adenocarcinoma; Stage: 37 I-II, 168 I-IIIA, 15 ‘advanced’, 85 NR.	Type: aerobic (HIIT), resistance, respiratory (IMT). FITT: ×3–5/week, 20–60 min, 2–20 weeks. Intensity: HIIT 80–95% iPPO (patient’s peak power)/60–100% Wpeak/70–80% 6 MWT speed/80% VO_2 peak_/RPE 15–17; resistance: 6–12 RM (reported in only 1 study); IMT: 50% Pimax/PEmax (reported in only 1 study). Supervision: All supervised.	Exercise only	Usual care (standard medical Rx, routine post-Rx physio, general info and monitoring from hospital)	Cardiorespiratory fitness (VO_2 peak_)	English and non-English (if a translation was available)
Zhou 2021 [42] Effects of exercise on fatigue in lung cancer. Inception—Mar 2020	8 RCTs (2× post-op, 1× post-op/post chemo, 2× during chemo, 2× during chemoRT, 1× during chemo or RT or targeted therapy, 1× during targeted therapy)	570 (283 post-op, 17 post-op +/− post adjuvant chemo, 111 chemo or RT or targeted therapy, 15 chemoRT, 120 chemo, 24 targeted therapy)	Type: 430 NSCLC, 140 NR. Stage: 228 I-IIIA, 179 III-IV, 163 NR.	Type: aerobic, resistance, balance, Tai Chi. FITT: ×1–5/week, 20–60 min (1 RCT commenced @ 5 mins), 6–12 weeks. Intensity: 1×RCT light, 1×RCT moderate, 1×RCT mod-high, 5×RCT NR. Supervision: 1×RCT supervised, 2×RCT home-based.	Behaviour support and education (×1 trial)	Usual care/conventional physio, general health education materials (×1 trial), daily stretching (×1 trial)	Fatigue *, depression, anxiety, HRQoL, functional capacity	English
During and/or post-treatment (non-surgical)								
Peddle-Mc Intyre 2019 [43] Exercise training for advanced lung cancer. Inception—July 2018	6 RCTs (all non-surgical–3× during Rx, 2× palliative Rx or scheduled/eligible for Rx, 1× post Rx)	221 (48 chemotherapy, 24 EGFR inhibitors, 11 unspecified treatment, 27 no treatment, 111 NR)	160 NSCLC/SCLC, 24 NSCLC, 27 NSCLC/mesothelioma, 11 NR; Stage 187 IIIA-IV, 27 ‘advanced’, NR 8.	Type: aerobic, resistance, respiratory (including IMT) FITT: 30–65 min, 1–5 days per week over 6–12 weeks. Intensity: 55–75% HRR, 60–80% VO_2 peak_ (RPE 11–17), 60–70% max HR, 40–70% PImax. Supervision: supervised +/− home-based.	15–20 min/week behavioural support sessions (×1 trial)	Usual care (including conventional physiotherapy). Qigong for 6 weeks (×1 trial)	Exercise capacity *, muscle strength, HRQoL, dyspnoea, fatigue, anxiety and depression, lung function, physical activity, adverse events, overall survival, performance status	No language restriction
Lee 2021 [44] Exercise intervention effects for people with lung cancer during chemotherapy. Jan 2000–May 2020	6 RCTs (all during chemotherapy Rx)	244, receiving chemotherapy (both radical and palliative)	201 NSCLC or SCLC, 43 NSCLC; Stage NR.	Type: aerobic, resistance, respiratory FITT: 20–75 mins, 2–5× per week over 6–12 weeks. Intensity: 60% HRR, 80–95% incremental peak power output, 30–80% peak work rate and 40–70% 1RM. Supervision: supervised and home-based.	Exercise only	Usual care	Pulmonary function, quality of life, pain, exercise capacity, strength, anxiety and depression, physiologic measurements (BP, max and resting HR, WBC, RBC)	English

**Footnote:** Colors represent timepoints: peach=Pre-treatment only (surgical); mustard=Post-treatment only (surgical); green=Pre and post-treatment (surgical); blue=Pre, during and/or post-treatment (surgical and non-surgical); purple=During and/or post-treatment (non-surgical). * denotes primary outcome (if specified). Abbreviations: 6 MWT = six-minute walk test; AEs = adverse events; AT = aerobic training; BP = blood pressure; COPD = chronic obstructive pulmonary disease; CPET = cardiopulmonary exercise testing; DLCO = diffusing capacity for carbon monoxide; FEV1 = forced expiratory volume in 1 s; FITT = frequency, intensity, type, time; FVC = forced vital capacity; HIIT = high intensity interval training; HR = heart rate; HRmax = maximum heart rate; HRQoL = health-related quality of life; HRR = heart rate reserve; ICC = inter-costal catheter; IMT = inspiratory muscle training; IP = inpatient; LC = lung cancer; LOS = length of stay; MHR = maximum heart rate; MIP = maximum inspiratory pressure; NR = not reported; NSCLC = non-small cell lung cancer; OP = outpatient; PEmax = maximum expiratory pressure; PImax = maximum inspiratory pressure; PKPCT = Power as Knowing Participation in Change Tool; PNF = proprioceptive neuromuscular facilitation; POC = post-operative complications; PPC = post-operative pulmonary complication; PT = physiotherapy; PWC = peak work capacity; QoL = quality of life; qRCT = quasi-randomised controlled trial; RBC = red blood cell; RCT = randomised controlled trial; RM = repetition maximum; ROM = range of motion; RPE = rating of perceived exertion; Rx = treatment; SCLC = small cell lung cancer; SR = systematic review; TNM = tumour, nodes, metastasis; VO2 peak = peak oxygen uptake; WBC = white blood cell; Wmax = maximum work capacity; Wpeak = peak work capacity. Meta-analysis completion (number of systematic reviews): pre n = 8/8, post n = 4/4, pre and post n = 3/6, pre, during and/or post n = 8/10, during and/or post n = 2/2. Risk of bias tool use (number of systematic reviews): Cochrane Risk of Bias pre n = 7/8, post n = 4/4 (+Jadad n = 1), pre and post n = 5/6 (+Downs and Black n = 1), pre, during and/or post n = 8/10, during and/or post n = 2/2; PEDro pre n = 1/8, pre and post n = 1, pre, during and/or post n = 2/10.

##### Pre-Treatment Only (Surgical)

Eight SRs assessed the effects of exercise training delivered pre-operatively [4,16,17,18,19,20,21,22]. Sample sizes ranged from 203 to 791 per SR. Exercise training was commonly multi-modal combinations of aerobic and resistance training. Two SRs focused on the effects of respiratory exercises [20] and inspiratory muscle training (IMT) [22]. Three SRs included exercise-only interventions whilst those remaining included RCTs where exercise was delivered as part of pulmonary rehabilitation or with nutrition, stress management and relaxation, psychological education or pharmacotherapy interventions. Program duration ranged from one to eight weeks. The reported primary outcome of interest was PPCs in four of the SRs, unspecified in two SRs, and exercise capacity and HRQoL in one SR, respectively.

##### Post-Treatment Only (Surgical)

Four SRs focused on exercise interventions delivered in the post-operative period [23,24,25,26]. Sample sizes ranged from 262 to 4381 participants per SR. One focused specifically on respiratory physiotherapy interventions delivered alone or combined with aerobic and resistance training [26]. Interventions were heterogeneous in the other three (combinations of aerobic, resistance, IMT, breathing exercises, balance and whole body vibration). Program duration ranged from 5 days [26] supervised in the inpatient setting to 20 weeks [24] supervised in the inpatient or outpatient setting combined with unsupervised home exercise. The primary outcome was exercise capacity in two SRs [24,25], HRQoL in one SR [23] and unspecified in the fourth SR.

##### Pre and Post-Treatment (Surgical)

Six SRs reported on the effects of exercise interventions pre- and post-surgery [27,28,29,30,31,32]. The sample sizes of included reviews ranged from 157 to 2068 participants per SR. Exercise interventions in all SRs included combinations of aerobic, resistance and respiratory training. The four most recent SRs included high-intensity interval training (HIIT) [29,30,31,32] and three SRs also included balance training [30,31,32]. One SR focused on the effect of respiratory exercise [29]. Program duration ranged from 5 days to 8 weeks pre-treatment and 5 days to 20 weeks in the post-treatment period. Outcomes of interest included exercise capacity, lung function, complications (including pulmonary), hospital length of stay (LOS), fatigue, HRQoL and mortality. The primary outcome was not specified by three SRs and was pulmonary function, HRQoL and PPCs/hospital LOS in the other three, respectively.

##### Pre, during and/or Post-Treatment (Surgical and Non-Surgical)

Ten SRs [33,34,35,36,37,38,39,40,41,42] synthesised the effects of exercise for people with lung cancer pre, during or following both surgical and non-surgical treatment. Sample sizes ranged from 187 to 2109 participants per SR. One SR question specifically focused on the effects of respiratory exercises [33] and another on HIIT [41]. Interventions included within SRs were typically combined aerobic, resistance and respiratory training and six SRs also included Tai Chi [34,36,38,39,40,42]. The program duration ranged from 1 to 20 weeks. Several SRs reported on the effects of exercise on single outcomes, namely depressive symptoms [33], sleep quality [34], HRQoL [37] and cancer-related fatigue [40,42]. In the remaining SRs, the primary outcome/s of interest was not reported by four SRs and was dyspnoea and exercise capacity in one SR.

##### During and/or Post-Treatment (Non-Surgical)

Two SRs evaluated the effects of exercise during and following treatment only and included only non-surgical participants [43,44]. One SR included only participants receiving curative or palliative chemotherapy [44]. Sample sizes were 221 and 244 participants per SR. Both SRs included aerobic, resistance and respiratory exercise interventions of 6 to 12 weeks duration. Exercise capacity was reported as the primary outcome of interest in one SR [43] and was not reported by the other SR [44].

### 3.4. Intervention Effects

A GRADE evidence summary map is shown in Figure 2. Figure 3, Figure 4 and Figure 5 and Appendix A provide a summary of meta-analyses and GRADE evidence certainty for the primary outcomes of interest of this overview of SRs and the SR quality ratings (AMSTAR-2). Appendix A summarises details of the exercise interventions of these meta-analyses. Further details are provided narratively below.

**Figure 2 jcm-12-01871-f002:**
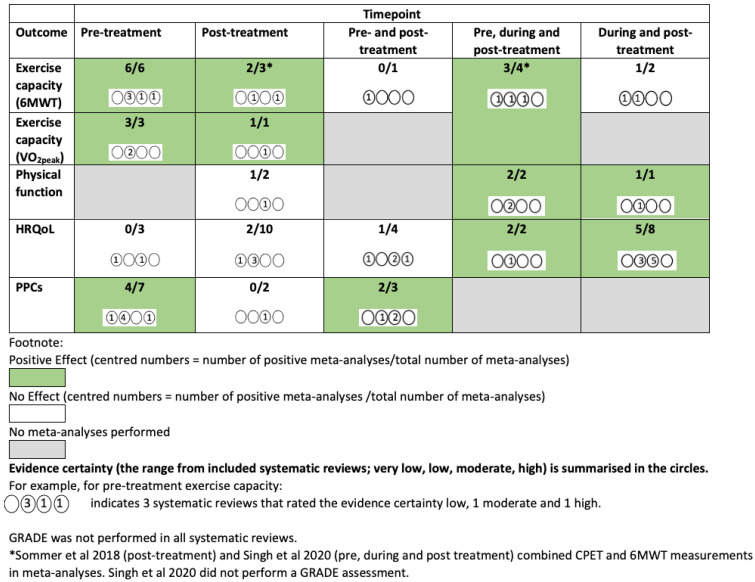
GRADE evidence map (excluding subgroup analyses and longer-term follow-up) [24,36].

#### 3.4.1. Pre-Treatment Only (Surgical)—Figure 3

##### Exercise Capacity

Six SRs performed meta-analyses of the effects of exercise training on exercise capacity measured by the 6 MWT [4,18,19,20,21,22] and three of these also analysed CPET VO_2 peak_ [18,19,20]. The sample size in meta-analyses ranged from 81 to 523. All meta-analyses reported significant positive effects of exercise training on exercise capacity, with mean differences (95% CI) ranging from 18.23 m (8.5, 27.96) [4] to 71.25 m (39.68, 102.82) [19]. The GRADE evidence certainty was ‘low’–‘high’. Gravier and colleagues performed a subgroup analysis according to intervention length, reporting greater effects for programs > 2 weeks (MD (95% CI) 95.89 m (27.15, 164.64) versus 25.9 m (14.61, 37.18)) [21].

**Figure 3 jcm-12-01871-f003:**
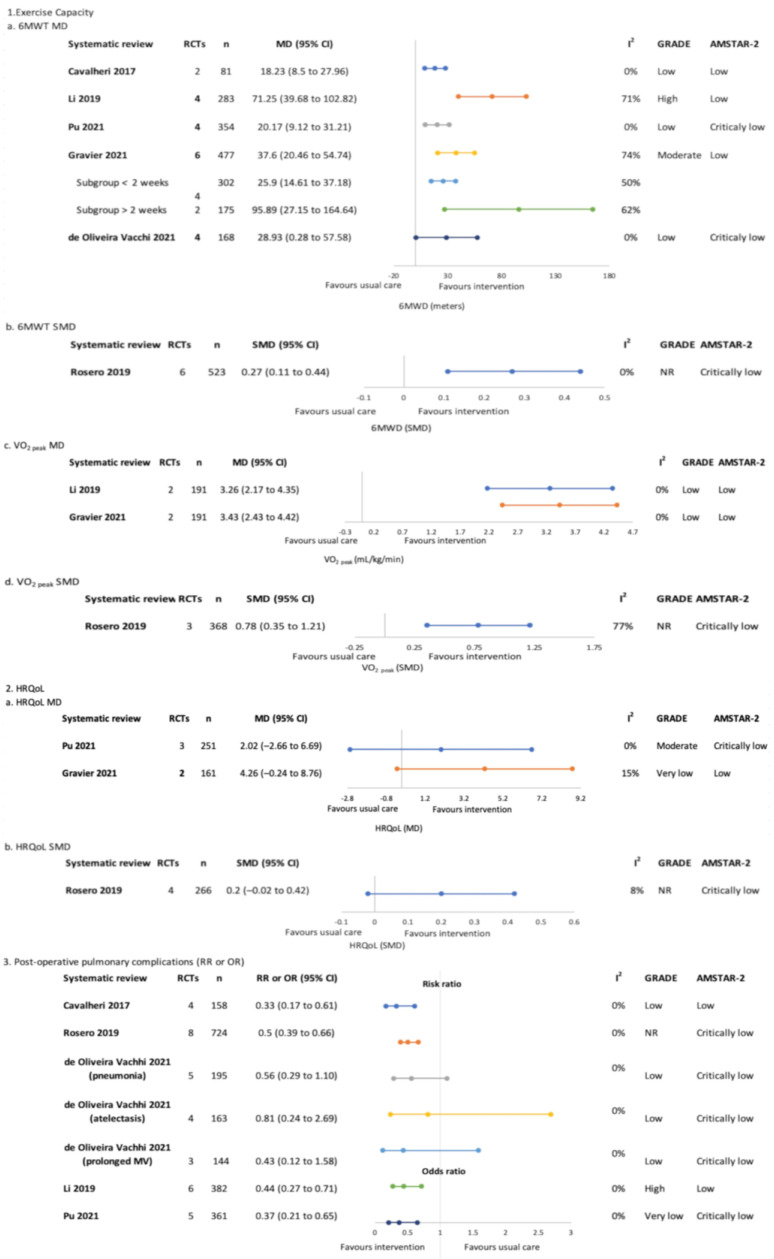
Pre-treatment only (surgical)–meta-analysis findings for overview primary outcomes (1. Exercise capacity, 2. HRQoL and 3. Post-operative pulmonary complications) [4,18,19,20,21,22].

##### Health-Related Quality of Life

Health-related quality of life was synthesised in the meta-analyses of three SRs [18,20,21], with sample sizes ranging from 161 to 266 participants. Non-significant between-group differences were reported in all meta-analyses. The GRADE evidence certainty was ‘very low’–‘moderate’.

##### Post-Operative Pulmonary Complications

Five SRs (including seven meta-analyses) assessed PPCs [4,18,19,20,22]. Sample sizes ranged from 144 to 724. Four meta-analyses were significant for reduction of odds or risk of PPCs favouring the intervention group [4,18,19,20]. The GRADE evidence certainty was ‘very low’–‘high’.

#### 3.4.2. Post-Treatment Only (Surgical)—Figure 4

##### Exercise Capacity

The positive effects of exercise training on exercise capacity (VO_2 peak_ or 6 MWT) were reported by two of three SRs immediately post-program [24,25]. This was not maintained at longer-term follow up (SMD 0.09 (−0.44 to 0.61, n = 56) [24]. Only one SR reported a subgroup analysis of rehabilitation timing, reporting no significant effects of exercise training on exercise capacity with early initiated rehabilitation, in contrast to interventions initiated later (SMD 0.58 (0.07, 1.09)) [24]. The GRADE evidence certainty was ‘low’–‘high’.

**Figure 4 jcm-12-01871-f004:**
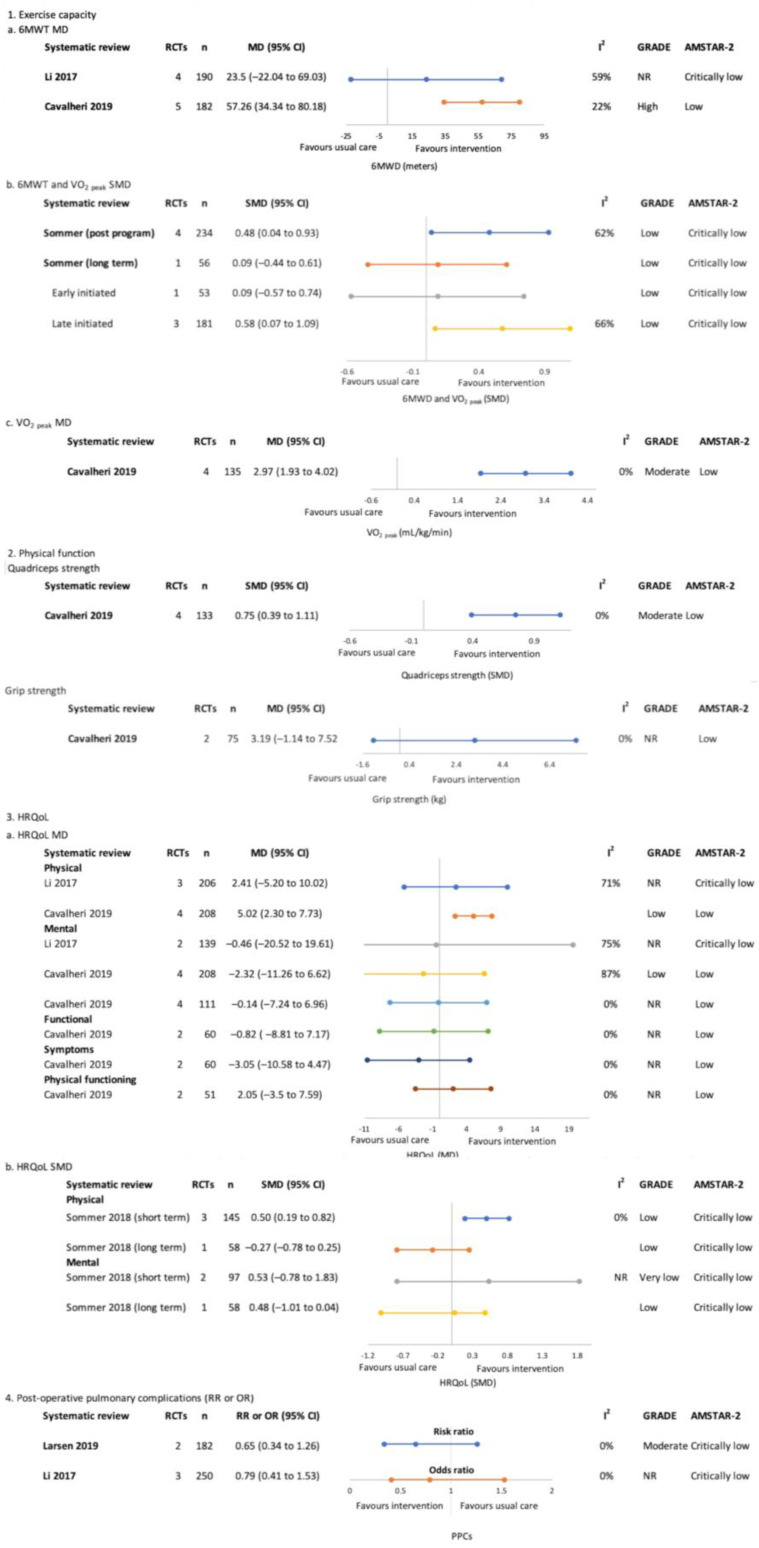
Post-treatment only (surgical)–meta-analysis findings for overview primary outcomes (1. Exercise capacity, 2. HRQoL and 3. Physical function and 4. Post-operative pulmonary complications) [23,24,25,26].

##### Physical Function

One SR reported significant effects of exercise training on quadriceps force-generating capacity (SMD 0.75 (0.39, 1.11), ‘moderate’ GRADE evidence certainty) and non-significant effects for handgrip force-generating capacity (MD 3.19 kg (−1.14, 7.52), GRADE evidence certainty NR) [25].

##### Health-Related Quality of Life

Three SRs [23,24,25] completed meta-analyses of HRQoL globally or by domain. The findings were non-significant aside from two meta-analyses of positive findings for HRQoL physical components immediately post-program (SMD 0.50 (0.19, 0.82) and MD 5.02 (2.30, 7.73)) which were not maintained in the one SR that assessed at 1-year follow-up (SMD 0.27 (−0.78, 0.25)) [24]. The GRADE evidence certainty was ‘very low’–‘low’.

##### Post-Operative Pulmonary Complications

Two SRs reported on post-operative complications with neither reporting significant effects of post-operative exercise, the GRADE evidence certainty was ‘moderate’ [23,26].

#### 3.4.3. Pre and Post-Treatment (Surgical)—Figure 5

##### Exercise Capacity

One SR focused on the effects of respiratory exercise with or without additional exercise components performed a meta-analysis for exercise capacity which demonstrated no significant between-group differences (MD 15.61 m (−24.05, 55.27), the GRADE evidence certainty was ‘very low’) [29].

**Figure 5 jcm-12-01871-f005:**
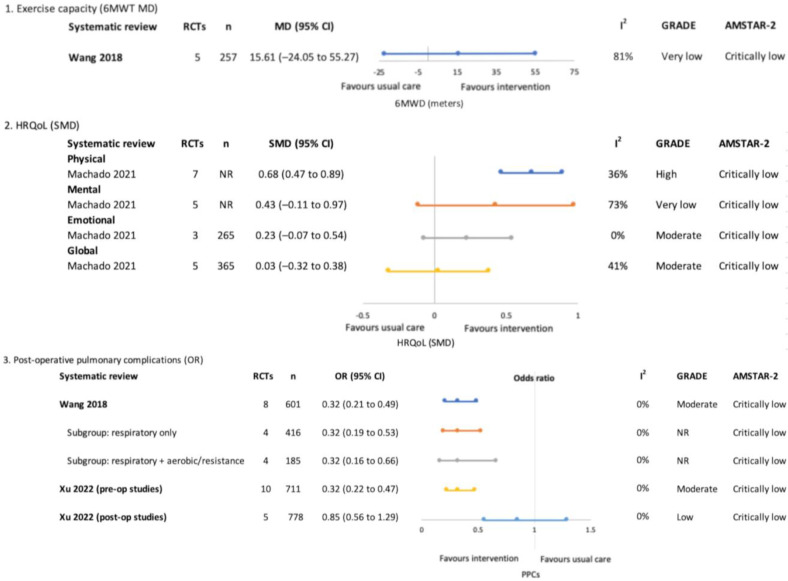
Pre and post-treatment (surgical)–meta-analysis findings for overview primary outcomes (1. Exercise capacity, 2. HRQoL and 3. Post-operative pulmonary complications) [29,31,32].

##### Health-Related Quality of Life

One SR meta-analysed HRQoL and reported significant effects favouring exercise interventions for the physical HRQoL domain (SMD 0.68 (0.47, 0.89), the GRADE evidence certainty was ‘high’) [31]. The findings for mental and emotional HRQoL domains and global HRQoL were not significantly different between groups.

##### Post-Operative Pulmonary Complications

Two SRs reported that exercise interventions reduced the odds of PPCs, with ‘moderate’ GRADE evidence certainty [29,32]. Wang and colleagues also performed a subgroup analysis according to exercise intervention type, both respiratory-only interventions (OR (95% CI) 0.32 (0.19, 0.53)) and respiratory combined with aerobic and resistance interventions (OR (95% CI) 0.32 (0.16, 0.66)) were reported to be effective [29]. One SR analysed pre- and post-treatment studies separately and reported positive effects only when exercise was performed pre-treatment (OR 0.32 (0.22, 0.47), GRADE evidence certainty was ‘moderate’) compared to post-treatment (OR 0.85 (0.56, 1.29), GRADE evidence certainty was ‘low’) [32].

#### 3.4.4. Pre, during and/or Post-Treatment (Surgical and Non-Surgical)—Appendix A

##### Exercise Capacity

Four meta-analyses synthesised the effects of exercise capacity [35,36,41,42]. Three of four reported significant increases in exercise capacity favouring the intervention (MD between 20.4 and 37.7 m for the 6 MWT) [35,36,41], the GRADE evidence certainty was ‘very low’–‘moderate’. In subgroup analyses, Singh et al. reported significant effects from meta-analyses involving aerobic-only and combined interventions and programs of <12 weeks or ≥12 weeks; both resulted in significant increases in exercise capacity compared to usual care. Subgroup analyses of supervised interventions were more effective than unsupervised (SMD 0.54 (0.32, 0.76) versus SMD 0.95 (−0.25, 2.16)) [36]. Both breathing exercises only and breathing exercises combined with aerobic and resistance significantly increased exercise capacity [35].

##### Physical Function

The positive effects of exercise interventions were found for upper and lower limb strength (SMD (95% CI) 0.59 (0.30, 0.88) and 0.38 (0.16, 0.61), respectively, GRADE evidence certainty evidence was ‘low’) [36]. Upper limb strength changes were no longer significant in subgroup analyses of aerobic exercise alone (SMD (95% CI) 0.14 (−0.23, 0.51)). Programs of <12 weeks or ≥12 weeks both resulted in significant increases in upper limb strength compared to usual care. For lower limb strength, subgroup analyses demonstrated that improvements were no longer significant when including only aerobic exercise, unsupervised programs, or programs <12 weeks duration (SMD (95% CI) 0.32 (−0.03, 0.66), 0.28 (−0.72, 1.28) and 0.28 (−0.10, 0.66)), respectively [36].

##### Health-Related Quality of Life

Two meta-analyses demonstrated the positive effects of exercise interventions for global HRQoL with ‘low’ GRADE evidence certainty [36,42] and these were maintained in subgroup analyses of exercise type (aerobic only or combined) and program duration (<12 weeks or ≥12 weeks). Positive effects remained for supervised programs (SMD 0.36 (0.24, 0.48) but not unsupervised (SMD −0.02 (−0.33, 0.30)) [36].

#### 3.4.5. During and/or Post-Treatment (Non-Surgical)—Appendix A

##### Exercise Capacity

Both SRs synthesised effects of exercise training on exercise capacity (measured by the 6 MWT), one reported positive effects (MD 63.33 m (3.70, 122.96)) and the other non-significant effects (SMD 0.38 (−0.42, 1.18)), the GRADE evidence certainty was ‘very low’–‘low’ [43,44].

##### Physical Function

Upper limb strength was reported to be significantly improved following exercise in one meta-analysis (SMD 1.39 (0.80, 1.98), the GRADE evidence certainty was ‘low’ [44].

##### Health-Related Quality of Life

Peddle-McIntyre et al. reported non-significant meta-analysis findings for physical HRQoL and positive findings favouring the intervention group for general HRQoL, GRADE evidence certainty was ‘low’ [43]. Lee reported significant benefits favouring the intervention group for physical, social, functional and general well-being, the GRADE evidence certainty was ‘moderate’ [44].

#### 3.4.6. Safety

One SR evaluated the safety of exercise interventions in the post-operative population. Four of eight of the included RCTs reported on adverse events, with only a single serious adverse event occurring [25]. In the one meta-analysis performed pre, during and/or post treatment, there were no significant differences in adverse events (grade 3–5 CTCAE severity ratings) between intervention and usual care participants (32 RCTs, n = 2109, 64 events (intervention) versus 61 events (usual care), risk difference −0.01 (−0.02, 0.01), I^2^ = 17%). The differences remained non-significant for subgroup analyses of exercise type (aerobic only, resistance only, combined or other), supervised/unsupervised and program duration (<12 weeks or ≥12 weeks) [36]. Data regarding the safety of exercise during and/or following treatment in people with lung cancer managed non-surgically was limited. One SR reported on adverse events and supported the safety of exercise training in advanced lung cancer with no serious adverse events (e.g., mortality, fractures) and limited minor adverse events (musculoskeletal injuries) occurring [43].

### 3.5. Overlap of Included Systematic Reviews

The corrected covered area calculation was 4.9% (see Appendix A in the Appendix A). This represents a slight overlap of primary studies (RCTs) included within this overview.

## 4. Discussion

This overview of reviews synthesised findings from 30 systematic reviews of over 6000 participants and investigated the efficacy and safety of exercise for people with surgical and non-surgical lung cancer across the care continuum. AMSTAR-2 ratings of the included systematic review quality were predominantly ‘very low’ to ‘low’, highlighting areas for improvement in future systematic review conduct and reporting. Clear efficacy exists for exercise interventions in lung cancer surgical populations, the minority of those diagnosed, particularly with respect to the prehabilitation period, for the outcomes of exercise capacity and post-operative pulmonary complications. It should be noted, however, that the GRADE certainty of evidence for these outcomes ranged from ‘low’ to ‘high’. In non-surgical lung cancer populations, additional higher-quality evidence is required to support the efficacy of exercise interventions. Only three systematic reviews, across both operable and inoperable populations, synthesised safety (adverse event) findings and all three reported few adverse events associated with exercise across the lung cancer care continuum. The need for improved transparency and consistency of reporting within studies is evident, with safety often not reported by the included RCTs.

Adding to the weight of evidence supporting lung cancer prehabilitation synthesised in this overview of reviews, the recently updated Cochrane systematic review, including 10 RCTs of over 600 participants, found high certainty evidence of a large reduction in post-operative pulmonary complication risk (RR (95% CI) 0.45 (0.33 to 0.61)) and moderate certainty evidence of an increase in exercise capacity (VO_2 peak_ MD 3.36 mL/kg/min (2.70 to 4.02)) associated with pre-operative exercise [45]. In the lung cancer surgical population, our attention needs to now focus on cost-effectiveness studies and the implementation of research to identify effective strategies for implementing exercise interventions into usual care. An excellent example of this is the UK-based ‘Prehab4Cancer’ lung cancer program which commenced in 2019 and services the greater Manchester area. Developed through a multi-disciplinary collaboration between clinical groups, a regional cancer alliance and community leisure centers, this community-based service includes exercise with nutrition and psychology also provided, dependent on screening criteria. In the 11 months prior to COVID-19, 377 people with lung cancer from 11 hospitals were referred. Seventy-four percent completed a baseline assessment and 48% completed the prehabilitation phase. The median program attendance was six sessions. Significant and clinically meaningful post-program improvements in objective (a 43 m increase in 6 MWT distance) and patient-reported (physical activity and HRQoL) outcomes were reported, and there were no adverse events recorded [46].

Within the systematic reviews included in this overview, there was significant heterogeneity in terms of the exercise modalities included in the interventions and elements of exercise prescription. This high degree of heterogeneity of interventions limited the ability of included systematic reviews to perform subgroup analyses according to intervention characteristics and delivery settings, a secondary aim of this overview. Further research, including network meta-analyses, is needed to establish the optimal exercise intervention features (including modality, intensity and duration) and identifying those most likely to benefit. Lu et al. will investigate in the lung cancer surgical population the effects of different types of exercise training on HRQoL, exercise capacity, lung function, adverse events and mortality in a Bayesian network meta-analysis [47]. A recent RCT post-treatment randomised 90 people with stage I-III lung cancer, and cardiorespiratory fitness lower than normative values, to 1 of 4 training groups (stretching attention control, aerobic, resistance or combined aerobic and resistance) [48]. The trial reported high intervention attendance (median 90%) and minimal loss to follow-up (10%). Post-program findings included improvements in exercise capacity (VO_2 peak_) in the aerobic and combined training groups compared to the attention control group. Muscle strength was also improved in the resistance and combined groups compared to the aerobic or attention control groups. It must be noted that only 56% (90/160) of the required sample size were recruited to this trial, resulting in a higher likelihood of statistical error, and findings should be interpreted with caution. The relative dose intensity, defined as the ratio of total ‘completed’ to total ‘planned’ exercise was higher in the aerobic training group, indicating that aerobic training may be more tolerable for survivors [48]. In line with previous findings in other cancer types [49], the meta-analyses of systematic reviews included in this overview reported exercise intervention effectiveness for supervised rather than unsupervised interventions [36]. Advances in the fields of real-time monitoring and reporting of exercise programs need to continue to support the fidelity of performance. This will facilitate patient-centred care for people with lung cancer, where preference is often for home or community-based programs [50], whilst ensuring programs are supervised to enhance effectiveness.

The overview protocol was developed and registered a priori and guided by a robust methodology which included a duplicate performance of all overview stages. Only the included evidence from study designs at lower risk of bias was included; RCTs and qRCTs. The decision to include all eligible systematic reviews was aligned with the overview aim of summarising the body of evidence but resulted in an overlap of primary studies included in the overview and potential double counting of outcome data. However, robust methods were used to assess and document the degree of primary study overlap of the included systematic reviews.

## 5. Conclusions

This overview has synthesised a large body of evidence and provides a clear understanding of the gaps in the current evidence base regarding exercise for people with lung cancer and directions for future research. The evidence synthesised in this overview supports lung cancer exercise interventions to reduce complications and improve exercise capacity in pre- and post-operative populations, and research should now focus on implementation. Additional higher-quality research is needed, particularly in non-surgical populations, including subgroup analyses to determine optimal exercise types and settings.

## Data Availability

No new data were created or analyzed in this study. Data sharing is not applicable to this article.

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
