# Peer review of "Exercise across the Lung Cancer Care Continuum: An Overview of Systematic Reviews"

_jcm, 2023, doi:10.3390/jcm12051871_

Round 1

Reviewer 1 Report

Dear Authors

Thank for giving me the opportunity to read this interesting manuscript. As you are aware of patients with lung cancer can divides into two groups. The group with curative options which is more or less 20%-30% and the group that do not have a curative option, the remaining 70%-80%. I would prefer that this was mentioned in the introduction so reader throughout the manuscript would have this knowledge. This means the reader would have an idea of survival in the respective groups you have defined. It could also be added in your discussion or conclusion that where we have the best evidence of the effect of exercise is in the minority of patients with lung cancer that have a curative option.

I do only have few comments to your manuscript.

On page 15 figure 2.

It was really difficult for me to understand figure 2.

I did get the effect part (green, white and grey), but when it comes to "evidence certainty" the grey colors confuses in the explaining text. Maybe just showing one circle for "very low" and two for "low" etc.

                        Low                                                  ⃝⃝                   very low

⃝⃝⃝              Moderate                                     ⃝⃝⃝⃝         high

And instead of colors in the "GRADE" part you could used numbers, so you would applied the numbers of studies instead of a color.

Alternatively you could described one square in detail for better understanding.

In your discussion line 493 you write:

“Despite being underpowered, post-program findings included significant improvements in exercise capacity (VO2 peak) in the aerobic and combined training groups compared to the attention control group. Muscle strength was also significantly improved in the resistance and combined groups compared to the aerobic or attention control groups. The relative dose intensity, defined as the ratio of total ‘completed’ to total ‘planned’ exercise was higher in the aerobic training group, indicating that aerobic training may be more tolerable in survivors [46].”

This hole manuscript is about evidence and certainty of evidence and with that in mind you highlight a study that have underpowering as a “huge” bias, they only include half of what they were powered to. This means that you cannot talk og significant results in each of 4 groups. So to be true to your manuscript you need to at least remove significant or mention that these results probably is due to type II errors.       

Best

Morten Quist

Reviewer 2 Report

Congratulations on the study carried out, it is certainly a relevant subject for the area of​​rehabilitation in oncology.

I will list below some considerations and suggestions for the manuscript.

Abstract

·       The authors could insert in the abstract (results), as a suggestion, some commentaries about the types of exercise intervention and delivery settings find across the review. However, this information is not fully described in the included reviews, and it would be important to highlight this lack of information.

Introduction

Suggestion:

The introduction is well described, but additional information about the safety and tolerability of exercise-based rehabilitation in patients undergoing lung cancer treatment may be of interest.

Additional information about exercise-based rehabilitation, physical therapy and breathing exercises may also be of interest.

Materials and Methods

There is no consideration about this.

Results

Check the resolution quality of the images.

Table 1:

Consider merging the first column with the second column and the third column with the fourth column.

Figure 2

The strategy used to describe the number of SR that used GRADE is not clear enough to facilitate the reading of this information. If there were a simpler way to separate information about evidence certainty and use of GRADE it would be interesting.

Line 298: (n=6) should be replaced by the citation of the studies that carried out the analysis in question. It would also be important to mention at the end of the first sentence the three studies that included the analysis of oxygen consumption. As a suggestion, it is important to review the text again, looking for situations where this type of adjustment is still necessary. Review articles bring with them many citations and this management is not a trivial task. See lines 307 and 308, for example. References must be at the end of line 308 and not on line 309.

Discussion

As suggested for the introduction, information on safety and tolerance, as well as the details of the interventions could be further explored in the discussion of the work as a way of promoting the search for a consensus regarding a minimal model of physical rehabilitation for lung cancer.

Conclusion

There is no consideration about this.

References

Check references 19, 20 and 30, authors' names are incorrect (some wrong characters are typed next to the authors' names).
